# Comparisons between Bioelectrical Impedance Variables, Functional Tests and Blood Markers Based on BMI in Older Women and Their Association with Phase Angle

**DOI:** 10.3390/ijerph19116851

**Published:** 2022-06-03

**Authors:** Rafael Oliveira, César Leão, Ana Filipa Silva, Filipe Manuel Clemente, Carlos Tadeu Santamarinha, Hadi Nobari, João Paulo Brito

**Affiliations:** 1Sports Science School of Rio Maior—Polytechnic Institute of Santarém, 2040-413 Rio Maior, Portugal; jbrito@esdrm.ipsantarem.pt; 2Life Quality Research Centre, 2040-413 Rio Maior, Portugal; 3Research Centre in Sport Sciences, Health Sciences and Human Development, Quinta de Prados, Edifício Ciências de Desporto, 5001-801 Vila Real, Portugal; anafilsilva@gmail.com; 4Escola Superior Desporto e Lazer, Instituto Politécnico de Viana do Castelo, Rua Escola Industrial e Comercial de Nun’Álvares, 4900-347 Viana do Castelo, Portugal; cleao@esdl.ipvc.pt (C.L.); filipe.clemente5@gmail.com (F.M.C.); 5Research Center in Sports Performance, Recreation, Innovation and Technology (SPRINT), 4960-320 Melgaço, Portugal; 6Instituto de Telecomunicações, Delegação da Covilhã, 1049-001 Lisboa, Portugal; 7Empresa Municipal “Esposende2000”, Município Esposende, 4740-204 Esposende, Portugal; ctsantamarinha@hotmail.com; 8Department of Motor Performance, Faculty of Physical Education and Mountain Sports, Transilvania University of Braşov, 500068 Braşov, Romania; 9Department of Exercise Physiology, Faculty of Educational Sciences and Psychology, University of Mohaghegh Ardabili, Ardabil 56199-11367, Iran; 10Faculty of Sport Sciences, University of Extremadura, 10003 Cáceres, Spain

**Keywords:** aerobic capacity, blood parameters, elderly, Fullerton tests, functional capacity, hemodynamic profile

## Abstract

The aim of the present study was to compare electrical bioimpedance variables, blood markers and functional tests based on Body Mass Index (BMI) in older women. Associations between Phase Angle (PhA) with functional tests and blood markers were also analyzed. A total of 46 independent elderly people participated in the study, and they were divided into four groups according to BMI values: Group 1 (G1, BMI < 25 kg/m^2^); Group 2 (G2, BMI > 25–30 kg/m^2^); Group 3 (G3, BMI > 30–35 kg/m^2^); Group 4 (G4, BMI > 35 kg/m^2^). In addition to the weight and height used to calculate the BMI, the following body composition variables were collected: fat mass (FM), fat-free mass, intracellular water (ICW), extracellular water (ECW), total body water (TBW) and PhA (50 kHz) through InBody S10 equipment. Functional capacity was assessed using the Fullerton battery of tests: arm-curl; chair-stand; 6 min walking test (6MWT); time up-and-go test (TUG); standing on one leg (SOOL) and take 10 foot-lines (10FL). The main results showed differences between groups in the tests: 6MWT, SOOL and 10FL between G1 vs. G3 and G2 vs. G3 (*p* < 0.05); ACT, AIC and AEC between G1 vs. G4 (*p* < 0.05); FM among all groups (*p* < 0.05). Negative correlations were found between PhA and the agility test in G1 (r = −0.848; *p* = 0.008) and G4 (r = −0.909; *p* = 0.005); PhA and chair-stand in G3 (r = 0.527; *p* = 0.044); PhA and forearm flexion in G3 (r = 0.641; *p* = 0.010) and G4 (r = 0.943; *p* = 0.001); PhA and 6MWT in G4 (r = 0.771; *p* = 0.042). This study found that there is a clear trend towards better functional capacities with better parameters of body composition. Although there were no differences between groups in PhA, associations were found between different functional tests with PhA, which reveals the importance of this variable as a marker of health status.

## 1. Introduction

Women are known to live longer than men [1]; however, they are also more likely to get sick [2]. In fact, the greater life expectancy is associated with comorbidities due to the aging process [3]. Furthermore, the process of menopause contributes to this effect, developing an estrogen deficiency, and women become more prone to impairment in physiological and cognitive functions than premenopausal women [4,5]. These alterations lead to a gradual loss of the body’s functional abilities, such as strength and balance [6], increasing the risk of sedentary behavior [7,8]. Indeed, a reduction of fat-free mass, particularly the appendicular, is closely related to aging and independently associated with worse health outcomes and disabilities [9,10,11], leading to frailty, reduced life expectancy in healthy subjects and impaired quality of life associated with weakness, alongside increased personal and healthcare costs [12,13]. Having a healthy functional fitness includes a better physiological level to safely and independently perform daily life activities without undue fatigue and embraces components such as lower and upper body resistance, lower and upper body flexibility, aerobic capacity and motor agility/balance [8]. Therefore, to mitigate the aging scenario, physical activity should be encouraged, being essential to ensure a good quality of life in older people [14].

Closely related to endocrine variations, the aging process also changes arterial stiffness, weakening the cardiovascular protective effect, which increases the prevalence of hypertension [15]. In turn, the serum cholesterol concentration also changes across the lifespan. For instance, from 20 years of age, women increase their LDL cholesterol concentrations progressively [16], reaching a plateau between the age of 60 and 70 years of age [17]. In opposition, the serum HDL cholesterol concentrations remain constant [18] in contrast to triglyceride concentrations, which increase throughout their lifetime and are always higher in those using estrogens [17]. After menopause, the LDL cholesterol level increases in opposition to the HDL cholesterol level [18]. Additionally, it seems that estrogen plays a protective role for cholesterol in women, since it was previously suggested that loss of endogenous estrogen increases LDL, which starts in the perimenopausal period and continues to increase until the age of 60 years, with such higher levels maintained in the following years [19].

Many prospective cohorts have reported that dyslipidemia (high levels of lipids in blood) can predict the risk of hypertension [20,21]. In fact, hypertension and dyslipidemia are well acknowledged as two main contributing risk factors of cardiovascular diseases [17,22].

Systolic and diastolic blood pressure have been found to be influenced by the components of body composition [11]. Evidence shows that the adiposity lodged in the abdominal area, as well as low fitness, has been associated with several diseases, including cardiovascular diseases [23,24], type 2 diabetes [25,26], cancer [25,27], dementia [28,29] and depression [30,31], and also with an increased risk of all-cause mortality independently of the body mass index (BMI) [32,33].

Regarding body composition assessment, several techniques have been used. For example, body fat percentage can be measured by underwater weighing (densitometry), dual energy X-ray absorptiometry (DEXA), bioelectrical impedance analysis (BIA) and magnetic resonance imaging (MRI). However, densitometry, DEXA and MRI are expensive, inconvenient for the participant and not feasible to conduct in the field because they require specialized equipment [34]. To measure body composition, many prediction equations are available to estimate body components as a function of resistance, reactance, anthropometric variables (weight and height), sex and age [35,36]. In the last decade, bioelectrical impedance analysis (BIA) has been widely used, since it is a noninvasive, inexpensive and portable method [35]. However, the results’ reliability is closely dependent on the chosen equation [36]. The bioimpedance method enables one to identify changes in bioelectrical characteristics of cells that impact health markers and functional body composition [37], especially since the bioelectrical impedance vector analysis (BIVA) does not depend on equations or body weight. Instead, it is based on raw BIA measures [38].

Briefly, BIVA uses resistance and reactance [39]. Resistance reflects the decrease in the voltage and, consequently, the conductivity through ionic solutions of intra- and extra-cellular water ,while reactance is related to the delay in the flow of current, which is measured as a phase shift, reflecting the capacitance of cell membranes and tissue interfaces [40]. Both measurements allow one to access Phase Angle (PhA).

Clinically, the most established impedance parameter has been the PhA, defined as the geometrical angular transformation of the ratio between reactance and resistance [36,41]. The BIA technique, through the phase angle (PhA), records the transient changes in the electrical properties of the tissues [8]. This parameter has been used for th diagnosis of malnutrition and clinical prognosis, both associated with changes in cellular membrane integrity and alterations in fluid balance [35]. In fact, positive associations have been shown between PhA and survival in patients with HIV-positive AIDS [41,42], with lung cancer [43], undertaking hemodialysis [44,45] and who are critically ill [36,46]. With increasing age, a decrease in the PhA values has been observed, which may suggest that it is an indicator of function and general health, not only an indicator of body composition or nutritional status [35]. Therefore, this method has been suggested as an important tool for evaluating the clinical outcome or to monitor the progression of a disease [35].

Considering factors such as age, BMI and sex as being primary determinants of the PhA and that only two studies considered BMI to analyze PhA [47,48], the purpose of this study was to compare bioelectrical impedance variables, functional tests and blood markers based on BMI in older women. In addition, the associations between PhA, functional tests and blood markers were analyzed.

## 2. Materials and Methods

### 2.1. Participants

Participants were recruited from a municipally owned corporation, “Esposende2000” (Esposende, Portugal), to characterize older people of the city. Initially, 89 persons were recruited, but some failed to complete all assessments. Thus, the present cross-sectional study included 46 volunteer independent older women. They were divided according to the BMI cut-offs [49]. Group 1 (G1, BMI 18.5–24.9 kg/m^2^), Group 2 (G2, BMI > 25–29.9 kg/m^2^), Group 3 (G3, BMI > 30–35 kg/m^2^) and Group 4 (G4, BMI > 35 kg/m^2^). This division was based on previous research that considered BMI as one of the PhA determinant factors [47,48].

All participants were volunteers who received medical clearance to participate in the study. Before providing written informed consent to participate, each participant was informed about the study’s goals and potential benefits. The ethics committee of the Polytechnic Institute of Santarém approved the study (approval number: 092019) in accordance with the recommendations of the World Medical Association’s Declaration of Helsinki of 1975, as revised in 2013, for human studies.

To participate in the study, all women had to be physically independent (without depending on any help to perform daily tasks), without any osteoarticular dysfunction or uncompensated chronic diseases that could interfere with the performance of the proposed tests and without heart problems.

### 2.2. Procedures

Before all assessments, it was guaranteed that participants did not practice any exercise nor ingested caffeine or alcohol in the previous 12 h. In addition, all assessments except for functional fitness tests were performed after a minimum of 8 h of fasting and after emptying the bladder, in the morning. They were performed in a clinical facility with an ambient temperature and relative humidity of 22–23 °C and 50–60%, respectively. To avoid measurement errors, all assessments were performed by the same researchers.

#### 2.2.1. Anthropometric Assessment

Weight and height measurements were collected with participants wearing light clothes and no shoes, through a scale with a stadiometer (SECA 220, Hamburg, Germany), to the nearest 0.01 kg and 0.1 cm, respectively. Then, BMI was calculated using the standard formula (BMI = body mass [kg]/height^2^ [m]).

#### 2.2.2. Bioelectrical Impedance Analysis Assessment

All participants underwent BIA assessment using multifrequency tetrapolar InBody S10 equipment (Model JMW140, Biospace Co, Ltd., Seoul, Korea), following the manufacturer’s comprehensive guidelines [50,51]. On the day before the assessment, all participants were requested to not perform strenuous exercise, to not change their eating habits and to drink water as usual. Right before the beginning of the assessment, all metallic objects were removed. Then, the contact points where the electrodes were placed on the skin were cleaned with ethyl alcohol and hydrophilic cotton, and the patients remained in the supine position, still and quiet for 10 min. Then and according to the manufacturer instructions, to assess the multi-segmental frequency analysis, eight electrodes were placed in the following eight tactile points: thumbs, middle fingers and ankles of both hands and feet. After the beginning of the assessment, a total of 30 impedance measurements were collected across the following frequencies: 1, 5, 50, 250, 500 and 1000 kHz for right and left arms, trunk and right and left legs. Furthermore, 15 reactance and PhA measurements were also collected by using three different frequencies (5, 50, 250 kHz) for each arm, leg and trunk.

After all data collection, total body water (TBW), extracellular water (ECW), intracellular water (ICW), body fat (FM), fat-free mass (FFM), soft lean mass (SLM), body cell mass (BCM, visceral fat (cm^2^), the ECW/TBW ratio and PhA were used for analysis. PhA at 50 kHz was also used based on previous reports [51,52]. Posteriorly, two other ratios were calculated: ECW/ICW and the ECW/BCM.

#### 2.2.3. Hemodynamic Assessment

For the measurement of systolic blood pressure (SBP), diastolic blood pressure (DBP) and resting heart rate (RHR), a digital sphygmomanometer, Omron Digital Blood Pressure Monitor HEM-907 (Omron Healthcare Europe BV, Matsusaka, Japan), was used. Measurements were collected for three trials in a sitting position, after 15 min rest and with the left arm in support, with 5 min intervals [53]. They were registered in two consecutive days, and the mean value was considered.

#### 2.2.4. Blood Collection

On the second day of data collection upon arrival at the laboratory, participants rested in a seated position for at least 5 min. Fast plasma glucose (GLU), triglycerides (TG), total cholesterol (TC), low-density lipoprotein (LDL) and high-density lipoprotein (HDL) were determined by a fasting (minimum of 8 h fasting and at most 12 h) venous blood analysis performed in the local health Centre through standard international laboratory methods. Venous blood samples were collected into one tube between 7:00 a.m. and 9:00 a.m. and at a minimum of 72 h since the last physical exercise session. Five milliliters were withdrawn from a prominent superficial vein in the antecubital space using a clean venous puncture with minimal stasis and placed in a tube containing a dipotassiumethylenediaminetetra-acetic acid (EDTA) as an anticoagulant and preservative. All samples were centrifuged at 3000 rpm for 15 min, and plasma or serum aliquots were stored at −80 °C until being assayed. Inter- and intra-assay coefficients of variation were <10% as determined in human plasma. Measurements of serum levels of GLU, TG, TC and HDL were determined by standard methods. The LDL was calculated using the following equation: LDL = TC − (HDL + TGL/5) [54]. The analyses were carried out using a biochemical auto-analyzer system (Dimension RxL Max Biochemistry Integrated System, Siemens Dade Behring, Marburg Gmbh), according to established methods in the literature, consistent with the manufacturer’s protocol.

#### 2.2.5. Hemodynamic and Blood Composite Score

Each hemodynamic and blood variable was transformed into z-score (with the specific value of each participant subtracted by the average of the group for the variable and the result divided by the standard deviation of the group) to calculate a composite score (sum of the z-scores of all tests).

#### 2.2.6. Functional Capacity Tests

The assessment of functional capacity was conducted using functional tests chosen from the battery of Fullerton Functional Fitness test [8,55]: the arm curl (number of repetitions of bicep curl performed during 30 s with 2.5 kg), the 30 s chair stand (number of repetitions with the arms folded across chest of chair stand during 30 s), the 6 min walking test (6MWT, number of meters walking as fast as possible without running) and time-up-and-go test (TUG, number of seconds of one chair stand, walking as fast as possible during 2.44 m, turning and returning to the sitting position).

The balance tests chosen from the Fullerton Advanced Balance Scale were standing on one leg (number of seconds lifting the preferred leg during a maximum of 20 s with the arms folded across chest, without touching the other leg) and 10 foot line test (walking along a line, placing one foot directly in front of the other, such that the heel and toe are in contact on each step forward during 10 footsteps, where the number of the interruptions is recorded) [56]. A z-score value was used in both tests.

Each functional test was transformed into a z-score (with the specific value of each participant subtracted by the average of group for the outcome and the result divided by the standard deviation of the group) to calculate a composite score (sum of the z-scores of all functional tests).

### 2.3. Statistical Analysis

In the present study, data were analyzed using the IBM Statistical Package for the Social Sciences for Windows, version 23.0 (IBM Corp., Armonk, NY, USA). Descriptive statistics (mean ± standard deviation) were used, and both normality and homogeneity were verified for all variables using the Shapiro–Wilk test and Levene test, respectively. Variables without normality were analyzed through the Kruskal–Wallis and Mann–Whitney tests (standing on one leg and 10-foot line), while the others were analyzed through an ANOVA One Way and Bonferroni adjustment Post Hoc for independent samples. The significance level considered for all comparisons and correlations was set for a *p* ≤ 0.05. Additionally, the control of the false discovery rate using the Benjamini–Hochberg procedure [57] was used following all steps described in the Cramer et al. [58] study, since multiple comparisons were made (see supplement).

Hedge’s g effect sizes were also calculated to determine the magnitude of pairwise comparisons, and the Hopkins’ thresholds for effect size statistics were used, as follows: ≤0.2, trivial; >0.2, small; >0.6, moderate; >1.2, large; >2.0, very large; >4.0, nearly perfect [59].

The relationship between continuous variables in each of the groups was also verified using the Pearson product-moment correlation coefficient (r), thus determining the magnitudes of the associations (r = 0.10–0.29: small; r = 0.30–0.49: moderate; r = 0.50–1.0: strong) [60]. Then, simple linear regressions were also performed to identify which variable could better explain PhA.

Since we had no chance to recruit more participants, the sample power of the post-hoc F-test family was calculated for α level = 0.05, effect size = 0.6, four groups and *n* = 46. It was shown that there was a 91.8% power level (actual power) for the analysis. In addition, another sample power calculation was made for a post-hoc correlation (bivariate normal model) for α level = 0.05 and effect size = 0.5, which revealed the need of a sample with *n* = 46 to achieve 95.3% of actual power. Finally, the last sample power calculation was made for a post-hoc linear multiple regression (fixed model, R^2^ deviation from zero) for α level = 0.05, effect size = 0.35, *n* = 46 and number of predictors = 1, which revealed 97.5% of actual power. All power calculations were performed by G-Power [61].

## 3. Results

The comparisons between groups are presented in Table 1. To avoid false significant differences, the false discovery rate using the Benjamini–Hochberg procedure was applied. It was found that body cell mall presented a *p* = 0.001, which was lower than the α_adj_ = 0.002. This means that all the remaining null hypotheses were rejected for weight, BMI, FCT, total body water, intracellular water, extracellular water, body fat, visceral fat, fat free mass and body cell mass, and the testing stopped. Consequently, 10-foot line, time-up-and-go and 6WMT were not considered significant. For better clarity, see Appendix A Table A1).

Specifically, weight was lowest for G1 when compared to all groups: G1 vs. G2 (*p* = 0.003; g = −1.98), G1 vs. G3 (*p* < 0.001; g = −3.59) and G1 vs. G4 (*p* < 0.001; g = −4.07). G2 presented lower values than G3 (*p* = 0.001; g = −1.63) and G4 (*p* < 0.001; g = −3.15). G3 also presented lower values than G4 (*p* < 0.001; g = −1.87). BMI was lowest for G1 when compared to all groups: G1 vs. G2 (*p* < 0.001; g = −3.89), G1 vs. G3 (*p* < 0.001; g = −6.71) and G1 vs. G4 (*p* < 0.001; g = −6.17). G2 presented lower values than G3 (*p* < 0.001; g = −2.96) and G4 (*p* < 0.001; g = −4.84). G3 also presented lower values than G4 (*p* < 0.001; g = −2.78).

There were no differences in hemodynamic and blood variables between groups.

Regarding functional tests, G1 presented higher values when compared with G3 for the functional test composite score (FTC, *p* = 0.002, g = 1.48). G2 presented higher values when compared with G3 for FTC (*p* = 0.010, g = 1.29).

Considering the electrical bioimpedance variables, G1 and G2 presented lower values when compared with G4 (*p* < 0.001, g = −2.10 and *p* = 0.014, g = −1.31, respectively) in total body water. G1 presented lower values when compared with G4 (*p* = 0.001, g = −1.95) in intracellular water. G1 presented lower values when compared with G3 (*p* = 0.042, g = −1.23) and G4 (*p* < 0.001, g = −2.22), while G2 also presented lower values than G4 (*p* = 0.003, g = −1.59), in extracellular water.

Body fat presented the lowest values in G1 when compared with G2 (*p* = 0.002, g = −1.89), G3 (*p* < 0.001, g = −4.33) and G4 (*p* < 0.001, g = −4.98). G2 presented lower values than G3 (*p* = 0.002, g = −2.01) and G4 (*p* < 0.001, g = −3.65). G3 also presented lower values than G4 (*p* < 0.001, g = 2.14).

Visceral fat presented lower values in G1 when compared with G2 (*p* < 0.001, g = −1.72), G3 (*p* < 0.001, g = −4.08) and G4 (*p* < 0.001, g = −6.75). G2 presented lower values than G3 (*p* < 0.001, g = −1.73) and G4 (*p* < 0.001, g = −3.01). G3 also presented lower values than G4 (*p* = 0.017, g = −1.50).

Soft lean mass presented lower values in G1 and G2 when compared with G4 (*p* < 0.001, g = −1.98 and *p* = 0.017, g = −1.29, respectively).

Fat free mass presented lower values in G1 and G2 when compared with G4 (*p* = 0.001, g = −1.96 and *p* = 0.031, g = −1.20, respectively).

Body cell mass presented lower values in G1 and G2 when compared with G4 (*p* = 0.001, g = −1.93 and *p* = 0.041, g = −1.13, respectively).

Our sample power analysis showed that correlations for the whole group presented a good power analysis, which was not the case when we divided the analysis per groups. Considering the overall sample analysis (all participants), it was verified that all variables correlated with each other, with the exception of PhA (all with large effect). Table 2 presents correlations between PhA with BMI, body fat and visceral fat by each group and overall sample.

Table 3 presents correlations between PhA and hemodynamic, blood variables and functional tests by each group and the overall sample. Considering the overall sample, there were positive (moderate to large) correlations between PhA and chair-stand, arm-curl, 6MWT, SOOL and FTC. Additionally, there were negative (moderate to large) correlations between PhA and SBP and TUG. When analyzing by each group, few correlations between different variable were noted (see Table 3).

Table 4 presents a multilinear regression analysis for the overall sample, which was performed to verify which variable’s (SBP, Chair-stand, TUG, Arm Curl, 6MWT, 10-foot line, FTC) agreement with the correlation analysis could be used to better explain PhA. After controlling the false discovery rate, it was found that arm curl test presented a *p* = 0.001, which was lower than the αadj = 0.002. This means that all the remaining null hypotheses were rejected, and the testing stopped (see Appendix A Table A2). Considering our sample power analysis, only the TUG test could be used to predict phase angle.

## 4. Discussion

This study analyzed the associations between PhA, functional tests and blood markers in healthy older women. Additionally, we compared bioelectrical impedance variables, functional tests and blood markers based on BMI.

As mentioned in the introduction, to measure body composition, several techniques have been used to assess percentage body fat in controlled laboratory conditions. However, the BIA allows the assessment of the PhA, which has been used as a possible indicator of good cell health, reflecting cell membrane integrity and better cell function [40,62]. Currently, the literature already supports an association between PhA values and some diseases, as well as an increased mortality rate, even though it was not possible to define cut-off values at this moment [63].

It is possible to find some literature that points to the existence of correlations of PhA values with gender, age and BMI, with women often having lower PhA values compared to men [35]. Furthermore, a decrease with age of the PhA values is already established [64]. Additionally, and more importantly, there seems to be a direct relationship with the BMI values below 35 kg/cm^2^ [47,65].

In our study, even considering the body composition differences in our participants, we did not find significant correlations for the PhA values in the overall participants for weight, BMI, body fat and visceral fat. One possible reason to not find an overall association may be because G4 is composed by participants who exceed 35 kg/cm^2^. Above this value, the correlations are more contradictory, which may have been the case in our study. For that reason, we performed a more detailed analysis, which showed a positive relationship between the PhA and the BMI in G1. It was also possible to establish a negative relationship between the PhA and fat mass and visceral fat in G3.

However, when we compared the values of the four groups in our study with the values of Bosy-Westphal et al. [47], the PhA values of G1 and G3 were slightly higher than those found by Bosy-Westphal et al. in the age group above 70. In the same order, G2 and G3 presented higher values compared to the values found for the same age group of Bosy-Westphal et al. [47].

The PhA of the total sample of 5.34° ± 0.57° in our study is lower than the values reported by Buffa et al. in 39 Italian women aged between 70 and 79 years, namely 5.5° ± 0.8° [66]. In the same vein, Barbosa-silva et al. also found higher values in 152 US healthy participants over 70, namely 5.64° ± 1.02° [35]. Finally, Saragat et al. evaluated a group of 295 Italian women over 65 years of age and reported mean values of 5.9° ± 1.0°, higher than those of our participants. If we consider only the mean value found for the age group between 65 and 70 and 71 and 80 years, we observed even higher values (6.0° ± 0.8° and 6.1° ± 1.0°, respectively) [67].

Although we only have the average value of the Italian participants of both sexes, the Basile et al. study reported an average value of 5.1° ± 1.0°, lower than in our sample [68]. The subjects in the Basile et al. study had a lower mean age than in our study (less ~4 years), which, at these more advanced ages, may mean a difference in the maintenance of muscle mass. Other aspects that may have an influence include the caloric intake, which may lead to sarcopenia through a deficiency in protein intake and subsequent decreases in muscle protein synthesis. Chronic inflammation, which is highly prevalent in the elderly, is thought to also be a key factor in the genesis of sarcopenia [69]. It has also been demonstrated that high PhA is well related in non-institutionalized elderly subjects with high physical activity levels [40].

The differences observed may reflect variations in muscle strength and in the quantity of the muscle mass of the different participants [62], but it may also reflect differences in methodological level with the use of different devices to assess PhA [70]. Actually, in a study of healthy individuals from the same country, which may eventually correspond to identical aging in terms of lifestyle, Matias et al. found similar values of 5.29° ± 0.54° in 92 female participants [71].

Regarding the relationship between PhA and functional fitness, there are already several studies that suggest the possibility of using this value as an indicator of muscle mass and strength reduction [40,68]. Confirming this relationship, our results show a positive relationship between the PhA and the functional fitness tests performed, with the exception of the TUG test, which has a negative relationship but due to the fact that, in this test, a lower value corresponds to a better result.

Despite being healthy and autonomous participants, the values for the different functional fitness tests presented much higher mean values than those defined as the reference. When compared with the data defined for the mean age of our sample (71.1 years) by Rikli et al., we found that these were superior in all tests related to strength [72]. Despite the possible cultural differences (lifestyle habits) between different countries and populations used in the studies, the results in our study presented higher values than those found by Marques et al. [73] and those referred to as criterion data for the Portuguese population. Finally, when comparing with the Sardinha et al. study, and despite some discrepancies in participants’ age between both studies, we noticed higher values in all functional fitness tests of our participants [74]. In this sense, we can only speculate that the reason for these differences lies in the fact that the physical fitness levels of our participants were much higher than the physical fitness of the samples from the other groups. As mentioned above, the subjects of our sample performed physical activity with a weekly frequency of 2–3 times.

Although relationships between cardiovascular risk and PhA have already been described [75], we were only able to find a significant value of association between systolic blood pressure and PhA. This lack of relationship between cardiovascular risk variables and PhA in our study may be due to the fact that our participants had lower mean values than those observed by Saad et al. in systolic, diastolic, triglycerides, total cholesterol and LDL cholesterol [75]. In addition, the mean PhA value of our sample is much higher than that defined by Alves et al., of 4.8°, which would work as a cut-off point for the increased risk of suffering a cardiovascular event [76].

Despite the results found, it is important to highlight the limitations of our study. The main limitation is the small sample size, which included only female participants. As this is a cross-sectional study, it was not possible to determine cause and effect relationships. For this reason, it is important that more studies are carried out in order to determine the effectiveness of PhA as an indicator of functional capacity.

## 5. Conclusions

In conclusion, we showed a significant association between PhA and functional tests in older female adults, regardless of the potential confounding effects related to body composition. PhA can be a good indicator of muscle strength, upper and lower limbs, agility and balance in elderly women. In this sense, it will be another complementary tool to monitor changes in physical fitness related to the aging process.

## Figures and Tables

**Table 1 ijerph-19-06851-t001:** Descriptive (mean ± SD) and comparative statistics of anthropometric and biochemical characteristics between all groups for all variables.

Variables	Overall	G1 (*n* = 8)	G2 (*n* = 16)	G3 (*n* = 15)	G4 (*n* = 7)	Comparisons
Age (years)	71.1 ± 6.4	73.0 ± 8.1	70.5 ± 6.4	71.3± 5.9	70.0 ± 6.4	-
Weight (kg)	71.9 ± 12.8	56.0 ± 4.6 ^a,b,c^	66.9 ± 5.6 ^b,c^	76.5 ± 5.9 ^c^	91.7 ± 11.1	G1 < G2, G3, G4 G2 < G3, G4 G3 < G4
Height (cm)	153.9 ± 4.4	154.9 ± 5.4	153.7 ± 4.6	153.6 ± 4.2	154.1 ± 4.2	-
BMI (kg/m²)	30.3 ± 4.9	23.3 ± 1.1 ^a,b,c^	28.3 ± 1.3 ^b,c^	32.4 ± 1.4 ^c^	38.5 ± 3.2	G1 < G2, G3, G4 G2 < G3, G4 G3 < G4
Hemodynamic and blood assessment						
RHR (bpm)	69.8 ± 10.9	70.6 ± 13.6	69.8 ± 10.2	68.9 ± 11.5	71.1 ± 10.3	-
SBP (mmHg)	123.2 ± 14.9	125.0 ± 10.9	119.4 ± 15.0	128.3 ± 15.0	118.4 ± 17.5	-
DBP (mmHg)	70.0 ± 10.3	72.3 ± 6.3	68.0 ± 8.4	67.2 ± 10.2	78.3 ± 14.5	-
Glucose (mg/dL)	107.6 ± 27.4	110.0 ± 17.0	110.0 ± 31.1	103.3 ± 29.4	108.0 ± 26.5	-
Triglycerides (mg/dL)	112.5 ± 48.5	73.0 ± 7.8	106.5 ± 39.1	111.7 ± 53.8	158.3 ± 56.0	-
TC (mg/dL)	184.7 ± 29.2	159.0 ± 15.4	186.5 ± 25.8	183.4 ± 31.6	201.5 ± 11.1	-
HDL (mg/dL)	59.0 ± 12.2	57.7 ± 14.6	57.8 ± 12.6	62.8 ± 12.4	57.5 ± 13.5	-
LDL (mg/dL)	106.9 ± 26.0	91.6 ± 44.7	107.4 ± 31.4	107.8 ± 22.4	112.2 ± 9.7	-
HBC (A.U.)	−0.07 ± 2.9	−1.1 ± 4.0	−0.5 ± 2.5	0.02 ± 2.7	1.7 ± 2.5	-
Functional tests						
Chair-stand (reps)	25.4 ± 6.6	28.8 ± 9.5	26.8 ± 6.2	22.8 ± 4.2	24.0 ± 6.4	-
Time-up-and-go (s)	4.7 ± 1.0	4.4 ± 0.9	4.4 ± 0.6	4.8 ± 0.8	5.7 ± 1.4	G1 < G4 G2 < G4
Arm Curl (reps)	26.9 ± 5.4	28.6 ± 6.4	28.7 ± 5.4	25.3 ± 4.8	24.9 ± 4.8	-
6MWT (m)	612.8 ± 108.3	675.4 ± 107.3	648.8 ± 80.4	542.7 ± 123.1	599.4 ± 56.3	G1 > G3 G2 > G3
10-foot line (nr)	−0.02 ± 1.0	0.7 ± 0.4	0.3 ± 0.6	−0.8 ± 1.2	0.0 ± 0.9	G1 > G3 G2 > G3
SOOL (s)	−0.002 ± 1.0	0.7 ± 1.1	0.2 ± 1.0	−0.5 ± 0.7	−0.4 ± 1.2	-
FTC (A.U.)	−0.03 ± 3.4	2.5 ± 4.0 ^b^	1.0 ± 2.5 ^b^	−2.5 ± 2.8	−0.1 ± 2.6	G1 > G3 G2 > G3
Electrical bioimpedance assessment						
Phase Angle (º)	5.4 ± 0.6	5.2 ± 0.6	5.6 ± 0.6	5.1 ± 0.5	5.4 ± 0.6	-
Total Body Water (L)	30.1 ± 3.1	27.4 ± 2.0 ^c^	29.7 ± 2.5 ^c^	30.5 ± 2.6	33.5 ± 3.4	G1 < G4 G2 < G4
Intracellular Water (L)	18.4 ± 1.9	16.7 ± 1.2 ^c^	18.2 ± 1.6	18.5 ± 1.5	20.3 ± 2.2	G1 < G4
Extracellular Water (L)	11.7 ± 1.2	10.7 ± 0.8 ^b,c^	11.4 ± 1.0 ^c^	11.9 ± 1.0	13.2 ± 1.3	G1 < G3, G4 G2 < G4
Body fat (kg)	31.1 ± 9.9	18.8 ± 2.9 ^a,b,c^	26.6 ± 4.4 ^b,c^	35.3 ± 4.0 ^c^	46.7 ± 7.1	G1 < G2, G3, G4 G2 < G3, G4 G3 < G4
Visceral fat (cm^2^)	161.6 ± 53.3	88.8 ± 18.6 ^a,b,c^	139.4 ± 31.9 ^b,c^	191.7 ± 26.7 ^c^	230.8 ± 21.1	G1 < G2, G3, G4 G2 < G3, G4 G3 < G4
Soft lean mass (kg)	38.5 ± 3.9	35.1 ± 2.6 ^c^	38.0 ± 3.2 ^c^	39.0 ± 3.3	42.8 ± 4.4	G1 < G4 G2 < G4
Fat free mass (kg)	40.8 ± 4.1	37.3 ± 2.7 ^c^	40.3 ± 3.4 ^c^	41.2 ± 3.5	45.0 ± 4.6	G1 < G4 G2 < G4
Body cell mass (kg)	26.3 ± 2.7	24.0 ± 1.8 ^c^	26.1 ± 2.3 ^c^	26.5 ± 2.2	29.1 ± 3.1	G1 < G4 G2 < G4
ECW/ICW	0.6 ± 0.02	0.6 ± 0.02	0.6 ± 0.02	0.6 ± 0.01	0.7 ± 0.03	-
ECW/BCM	0.4 ± 0.01	0.4 ± 0.01	0.4 ± 0.02	0.4 ± 0.01	0.5 ± 0.02	-
ECW/TBW	0.4 ± 0.008	0.4 ± 0.007	0.4 ± 0.008	0.4 ± 0.005	0.4 ± 0.001	-

RHR, resting heart rate; SBP, systolic blood pressure; DBP, diastolic blood pressure; TC, total cholesterol; HDL, high-density lipoprotein; LDL, low-density lipoprotein; TBW, total body water; ECW, extracellular water; ICW, intracellular water; BCM, Body Cell Mass; (º), degree; L, litre; kg, kilogram; 6MWT, six minutes walking test; SOOL, standing on one leg. HBC, hemodynamic and blood test composite score; FTC, functional test composite score. ^a^ denotes the difference from G2; ^b^ denotes the difference from G3; ^c^ denotes the difference from G4; all, *p* < 0.05.

**Table 2 ijerph-19-06851-t002:** Correlation analysis between weight, BMI and body composition variables for each group and overall sample.

Measures	BMI	Body Fat	Visceral Fat	Phase Angle	Soft Lean Mass	Fat Free Mass	Body Cell Mass
**G1**							
Weight	0.519	**0.830 (*p* = 0.011)**	**0.760 (*p* = 0.029)**	0.478	**0.820 (*p* = 0.013)**	**0.808 (*p* = 0.015)**	**0.841 (*p* = 0.009)**
BMI		**0.757 (*p* = 0.030)**	0.694	**0.801 (*p* = 0.017)**	0.098	0.075	0.178
Body fat			**0.984 (*p* < 0.001)**	0.621	0.362	0.342	0.405
Visceral fat				0.529	0.259	0.242	0.296
Phase Angle					0.174	0.150	0.286
Soft lean mass						**0.999 (*p* < 0.001)**	**0.991 (*p* < 0.001)**
Fat free mass							**0.989 (*p* < 0.001)**
**G2**							
Weight	**0.705 (*p* = 0.002)**	**0.796 (*p* < 0.001)**	**0.636 (*p* = 0.008)**	−0.026	**0.637 (*p* = 0.008)**	**0635 (*p* = 0.008)**	**0.605 (*p* = 0.013)**
BMI		**0.848 (*p* < 0.001)**	**0.810 (*p* < 0.001)**	−0.367	0.079	0.082	0.04
Body fat			**0.964 (*p* < 0.001)**	−0.295	0.041	0.039	0.007
Visceral fat				−0.457	−0.178	−0.181	−0.224
Phase Angle					0.334	0.333	0.442
Soft lean mass						**>0.999 (*p* < 0.001)**	**0.992 (*p* < 0.001)**
Fat free mass							**0.992 (*p* < 0.001)**
**G3**							
Weight	**0.737 (*p* = 0.002)**	**0.819 (*p* < 0.001)**	**0.617 (*p* = 0.014)**	−0.472	**0.745 (*p* = 0.001)**	**0.745 (*p* = 0.001)**	**0.708 (*p* = 0.003)**
BMI		**0.744 (*p* = 0.001)**	**0.624 (*p* = 0.013)**	−0.358	0.396	0.387	0.362
Body fat			**0.947 (*p* < 0.001)**	**−0.653 (*p* = 0.008)**	0.228	0.228	0.177
Visceral fat				**−0.658 (*p* = 0.008)**	−0.053	−0.054	−0.100
Phase Angle					−0.042	−0.043	0.028
Soft lean mass						**0.999 (*p* < 0.001)**	**0.996 (*p* < 0.001)**
Fat free mass							**0.996 (*p* < 0.001)**
**G4**							
Weight	**0.904 (*p* = 0.005)**	**0.966 (*p* < 0.001)**	**0.892 (*p* = 0.007)**	0.086	**0.923 (*p* = 0.003)**	**0.920 (*p* = 0.003)**	**0.888 (*p* = 0.008)**
BMI		**0.965 (*p* < 0.001)**	**0.964 (*p* < 0.001**	−0.044	0.702	0.693	0.661
Body fat			**0.948 (*p* = 0.001)**	−0.105	**0.794 (*p* = 0.033)**	**0.787 (*p* = 0.036)**	0.742
Visceral fat				−0.018	0.699	0.689	0.667
Phase Angle					0.363	0.366	0.460
Soft lean mass						**>0.999 (*p* < 0.001)**	**0.994 (*p* < 0.001)**
Fat free mass							
**Overall**							
Weight	**0.945 (*p* < 0.001)**	**0.967 (*p* < 0.001)**	**0.898 (*p* < 0.001)**	−0.053	**0.803 (*p* < 0.001)**	**0.787 (*p* < 0.001)**	**0.774 (*p* < 0.001)**
BMI		**0.968 (*p* < 0.001)**	**0.923 (*p* < 0.001)**	−0.061	**0.636 (*p* < 0.001)**	**0.614 (*p* < 0.001)**	**0.607 (*p* < 0.001)**
Body fat			**0.968 (*p* < 0.001)**	−0.141	**0.624 (*p* < 0.001)**	**0.603 (*p* < 0.001)**	**0.589 (*p* < 0.001)**
Visceral fat				−0.201	**0.489 (*p* = 0.001)**	**0.467 (*p* = 0.001)**	**0.452 (*p* = 0.002)**
Phase Angle					0.173	0.175	0.264
Soft lean mass						**0.999 (*p* < 0.001)**	**0.994 (*p* < 0.001)**
Fat free mass							**0.995 (*p* < 0.001)**

Correlations in bold were significant.

**Table 3 ijerph-19-06851-t003:** Correlation analysis between PhA with hemodynamic, blood and functional tests for each group and overall sample.

Measures	PhA (G1)	PhA (G2)	PhA (G3)	PhA (G4)	PhA (Overall)
RHR	0.340	0.122	−0.009	0.700	0.205
SBP	−0.114	−0.439	−0.121	−0.210	**−0.307 (*p* = 0.038)**
DBP	−0.026	−0.106	0.186	−0.701	−0.131
Glucose	**1.000 (*p* = 0.01)**	0.628	0.090	0.472	0.406
Triglycerides	−0.908	**−0.648 (*p* = 0.043)**	−0.049	0.282	−0.035
TC	−0.105	−0.010	−0.494	−0.848	−0.115
HDL	0.267	0.182	−0.558	−0.772	−0.226
LDL	**−1.000 (*p* = 0.01)**	0.079	−0.520	−0.222	−0.210
HBC	0.283	−0.082	−0.261	−0.222	−0.058
Chair-stand	0.408	0.204	**0.527 (*p* = 0.044)**	0.628	**0.385 (*p* = 0.008)**
TUG	**−0.848 (*p* = 0.008)**	−0.365	−0.476	**−0.909 (*p* = 0.005)**	**−0.544 (*p* < 0.001)**
Arm Curl	0.423	0.239	**0.641 (*p* = 0.010)**	**0.943 (*p* = 0.001)**	**0.492 (*p* = 0.001)**
6MWT	0.496	0.246	0.028	**0.771 (*p* = 0.042)**	**0.314 (*p* = 0.036)**
10-foot line	−0.485	0.257	0.229	0.633	0.266
SOOL	−0.436	0.490	0.370	0.700	**0.311 (*p* = 0.035)**
FTC	0.027	0.418	0.415	0.732	**0.366 (*p* = 0.012)**

PhA, phase angle; RHR, resting heart rate; SBP, systolic blood pressure; DBP, diastolic blood pressure; TC, total cholesterol; HDL, high-density lipoprotein; LDL, low-density lipoprotein; TUG, time-up-and-go; 6MWT, six minutes walking test; SOOL, standing on one leg. HBC, hemodynamic and blood test composite score; FTC, functional test composite score. Correlations in bold were significant.

**Table 4 ijerph-19-06851-t004:** Values of the regression analysis explaining phase angle values for th overall sample.

Variables	B	SE of B	R	R^2^	Adjusted R^2^	F	*p*	Effect Size F^2^
SBP	−0.012	0.06	0.307	0.094	0.074	4.588	0.038 *	0.10
Chair-stand	0.035	0.012	0.385	0.148	0.129	7.646	0.008 *	0.17
TUG	−0.324	0.075	0.544	0.296	0.280	18.487	<0.001 *	0.42
Arm Curl	0.054	0.014	0.492	0.243	0.225	14.089	0.001 *	0.32
6MWT	0.002	0.001	0.314	0.099	0.078	4.701	0.036 *	0.11
10-foot line	0.155	0.085	0.266	0.071	0.050	3.359	0.074	0.08
FTC	0.064	0.025	0.366	0.134	0.115	6.822	0.012 *	0.15

Note: SE, standard error; confidence interval; * Significant differences at the 0.05 levels.

## Data Availability

The data presented in this study are available on request from the corresponding author.

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
