# Peer review of "Comparisons between Bioelectrical Impedance Variables, Functional Tests and Blood Markers Based on BMI in Older Women and Their Association with Phase Angle"

_ijerph, 2022, doi:10.3390/ijerph19116851_

Round 1
Reviewer 1 Report
This study analyzed the associations between PhA, functional tests, and blood markers in healthy older women. Additionally, the authors compare bioelectrical impedance variables, functional tests, and blood markers stratified by BMI.
Comments:
Variables described in Table 1 are expected results because are covariates of BMI or presented as characteristics of the increase in BMI since the variable used to stratify these groups in G1 to G4 was BMI
The authors mention that the group of G4 was composed of participants who exceeded the 35 kg/cm2, and above this value, the correlations are more contradictory, it is suggested that this group be eliminated from the analysis or justify the reason for its inclusion, or the importance of this group remaining in the analysis, although the analyzes were focused on G1, the G4 group could deviate the reported findings.
Author Response
Comments and Suggestions for Authors
This study analyzed the associations between PhA, functional tests, and blood markers in healthy older women. Additionally, the authors compare bioelectrical impedance variables, functional tests, and blood markers stratified by BMI.
Comments:
Variables described in Table 1 are expected results because are covariates of BMI or presented as characteristics of the increase in BMI since the variable used to stratify these groups in G1 to G4 was BMI
Authors: We agree with the opinion of the reviewer. We kept all comparisons to confirm and standardize all results section.
The authors mention that the group of G4 was composed of participants who exceeded the 35 kg/cm2, and above this value, the correlations are more contradictory, it is suggested that this group be eliminated from the analysis or justify the reason for its inclusion, or the importance of this group remaining in the analysis, although the analyzes were focused on G1, the G4 group could deviate the reported findings.
Authors: Dear reviewer, correlations of G4 does seem to be contradictory. For instance, Table 2 showed that all significant correlations were positive and direct which confirms that with a higher BMI (35 kg/cm2) there are a tendency for higher values of body composition variables. Additionally, table 3 showed 3 correlations where it was found that higher distance covered in the 6MWT, higher number of repetitions of the arm curl and a lower value of the TUG (negative correlation) was associated with a higher PhA for G4.
Reviewer 2 Report
I'd like to thank the authors for addressing my concerns. There are still a few questions that need to be addressed.
- What was the FDR set to for the Benjamini Hochberg? How many tests did you control for?
- I did not realize this, but there are apparently 7 different regression analyses completed. Please apply the FDR to those as well.
- Also based on how the FDR is done, you calculate the q (i.e. FDR=20%, so for your 42 analyses q would be 0.004761905), then you find your lowest p-value and compare it to that, then you move to the next lowest p-value and compare it to 0.20/41, etc etc. So once you arrange all of your p-values, your p=0.011 might be say the 10th lowest p-value which would mean that the q is 0.00625 (0.20/32) and the if p>q then your analysis not significant.
Author Response
Comments and Suggestions for Authors
I'd like to thank the authors for addressing my concerns. There are still a few questions that need to be addressed.
Authors: Thank you
- What was the FDR set to for the Benjamini Hochberg? How many tests did you control for?
Authors: We controlled FDR is with the Benjamini-Hochberg procedure (Benjamini & Hochberg, 1995). The following steps described by Cramer et al. (2016) was adopted: “First, one sorts all p-values in ascending order, that is, with the smallest p-value first (see also Figure 2 for a visualization of the method). Next, one computes an adjusted α level, αadj. For the largest p-value, αadj equals α times the rank number of the largest p-value (3 in our example) divided by the total number of tests (also 3 in this example): 0.05*(3/3) = 0.05. For the middle p-value, αadj equals 0.05*(2/3) = 0.0333; for the smallest p-value, αadj equals 0.05*(1/3) = 0.01667. Next, one evaluates each p-value against these adjusted α levels, with the largest p-value evaluated first. Importantly, if the H0 associated with this p-value is rejected (i.e., p < αadj) then all testing ends, and all remaining tests are considered significant as well.” We completed the explanation on the manuscript as well.
Cramer, A. O., van Ravenzwaaij, D., Matzke, D., Steingroever, H., Wetzels, R., Grasman, R. P., Waldorp, L. J., & Wagenmakers, E. J. (2016). Hidden multiplicity in exploratory multiway ANOVA: Prevalence and remedies. Psychonomic bulletin & review, 23(2), 640–647. https://doi.org/10.3758/s13423-015-0913-5
Benjamini, Y., & Hochberg, Y. (1995). Controlling the false discovery rate: a practical and powerful approach to multiple testing. Journal of the Royal Statistical Society. Series B (Methodological), 57, 289-300
- I did not realize this, but there are apparently 7 different regression analyses completed. Please apply the FDR to those as well.
- Also based on how the FDR is done, you calculate the q (i.e. FDR=20%, so for your 42 analyses q would be 0.004761905), then you find your lowest p-value and compare it to that, then you move to the next lowest p-value and compare it to 0.20/41, etc etc. So once you arrange all of your p-values, your p=0.011 might be say the 10th lowest p-value which would mean that the q is 0.00625 (0.20/32) and the if p>q then your analysis not significant.
Authors: The procedures were revised and a supplement was provided to confirm all results (see appendix table A1). In addition, FDR was applied to the regression analysis and a supplement table was provided for better, accordingly appendix table A2. Thank you
Round 2
Reviewer 2 Report
I'd like to thank the authors for addressing my concerns.
This manuscript is a resubmission of an earlier submission. The following is a list of the peer review reports and author responses from that submission.
Round 1
Reviewer 1 Report
Rephrase Line 87…”enable to identify…”
Need a stronger justification of why using BIA and PhA
Line 110 “women independent volunteer” needs to be reworded
Explain why BMI was used to group people
Line 120 “physically independent”. Explain what that means
Line 139-140 “participant was asked”. Rephrase
Line 143-144 “without noise”. In the environment or they had to remain still and quiet?
Line 144-145 and 150-151. Why were those body points selected for BioImpedence measures. Explain.
Line 150 “ethnic”=ethnicity?
Line 188. Rephrase
Line 229 “significant”=significantly?
In table 1 explain the ECW terms
In the Discussion strengthen the reasons for using impedence measures. For an audience naïve to these measures the reasons for using are not compelling.
Line 350 Why ate these studies “superior” to others. That is a strong statement.
Line 368 “small sample”=small sample size?
Line 376 “work”. Rephrase
After reading I am not really sure what the authors have shown that is different to that already in the literature, especially given all the limitations of the study. I would like if they could share that in a clearer way so the impact of the work can be determined. Phrasing of some sections needs attention. Need to feel more convinced that the the study analyzed something novel which currently I do not.
Reviewer 2 Report
This study analyzed the associations between PhA, functional tests, and blood markers in healthy older women. Additionally, the authors compare by bioelectrical impedance variables, functional tests, and blood markers stratified by BMI.
The study is interesting; however, it presents several points to work on in the results section to strengthen and highlight the findings of the study and improve the quality of the manuscript.
Major comments:
Several of the variables described in Table 1 are expected results because are covariates of BMI or presented as characteristics of the increase in BMI since the variable that stratified the groups from G1 to G4 was BMI. Authors are suggested to justify what is the purpose of showing table 1, it is suggested only to keep the data of the Electrical bioimpedance assessment
For the results described in Table 1, it is suggested to include a column where the p-value of the global comparison is described by means of an ANOVA test, and later in the text to describe which variables gave differences by means of a post hoc test.
Tables 2 and 3 are interesting results, however, the authors do not describe these findings in the text, leaving the interpretation of the reader, it is important that the authors write a paragraph describing the results that introduce the findings. obtained in each table.
Table 4 does not describe for which variables the regression analyses shown were adjusted.
Reviewer 3 Report
Overall it's a well-written study. Below are my comments
Introduction
- Well written introduction, one minor thing to bring up is that estrogen plays a protective role for cholesterol in women
Methods
- Did you perform any a priori power analyses?
- How were participants recruited? Where were they recruited from?
- This was a very well written methodology. Were the post-hoc Mann-Whitney U tests conducted using Bonferroni corrections? Why not just use a Dunn's test?
- Did you correct for the sheer number of analyses you were conducting? Based on the p-values you've presented as long as you use a liberal correction test you shouldn't have any trouble.
- You should be clear that in your regression you included the entire sample
Results
I would strongly recommend making the table easier to read by including another column where you provide post-hocs (i.e. G1>G2, G3, G4, etc etc)
The results are quite a bit to comprehend. I'd suggest finding a more efficient way to present them.
What was the post-hoc power calculation for your analyses?
Discussion
Based on the results you have presented the discussion is adequate although there are significantly more limitations than what you have mentioned, one being the sheer number of tests that you conducted.
I think this study has significant value if the above problems can be addressed.
Round 2
Reviewer 2 Report
The authors have addressed the recommendations made to improve the comprehension of the manuscript.
Minor comments:
Indicate in each table what type of statistical tests were used to obtain the p values ​​or type of correlations performed.
Reviewer 3 Report
Dear authors,
Thank you for taking the time to respond to my comments. I believe you may have mis-understood some of my comments. I will list the ones that weren't correctly addressed below.
- In the results table 1, instead of presenting the p values, you should remove that column and just write G1>G2, G3, G4, etc etc if they're statistically significant. This way you don't need to add additional columns.
- The results are still very hard to digest. I would recommend reading some of Dr. Martin Seligman's work on how to write a succint results section. As someone who is focused on data science, I am personally getting overwhelmed with it. An example sentence for the first part of your results can be "Weight was lowest for G1 compared to all other groups, while G2 was lower than both G3 and G4 and G3 was lower than G4 only."
- You ran 32 uncontrolled analyses, which is basically a fishing expedition because if you run enough analyses there is a chance of a Type I error that you'll find something statistically significant. Therefore, you need to account for this by correcting for multiple analyses. The post-hoc Bonferroni correction is great, but for multiple analyses you should look at something like a Benjamini-Hochberg False Discovery Rate
- In your controlled analysis did you run 7 different regression models? But when calculating power you used it as a single model with 7 variables? Where did you get an effect size of 0.6?
There are still significant issues with the statistical analysis section and the results need to be re-written.
